



# Development of a versatile source apportionment analysis based on positive matrix factorization: a case study of the seasonal variation of organic aerosol sources in Estonia.

Athanasia Vlachou[1], Anna Tobler[1], Houssni Lamkaddam[1], Francesco Canonaco[1], Kaspar R. Daellenbach[1, †], Jean-Luc Jaffrezo[2], María Cruz Minguillón[3], Marek Maasikmets[4], Erik Teinemaa[4], Urs Baltensperger[1], Imad El Haddad[1] and André S.H. Prévôt[1]

[1]Department of General Energy Research, Paul Scherrer Institute, Villigen PSI, CH-5232, Switzerland

[2]Université Grenoble Alpes, CNRS, IRD, G-INP, IGE, 38000 Grenoble, France

[3]Institute of Environmental Assessment and Water Research (IDAEA), CSIC, 08034 Barcelona, Spain

[4]Estonian Environmental Research Centre, 10617, Tallinn, Estonia

[†]Now at: Institute for Atmospheric and Earth System Research / Physics, Faculty of Science, University of Helsinki, P. O. Box 64, 00014, Helsinki, Finland

*Correspondence to*: André S. H. Prévôt (andre.prevot@psi.ch) and Imad El Haddad (imad.el-haddad@psi.ch)

**Abstract.** Bootstrap analysis is commonly used to capture the uncertainties of a bilinear receptor model such as the positive matrix factorisation (PMF) model. This approach can estimate the factor related uncertainties and partially assess the rotational ambiguity of the model. The selection of the environmentally plausible solutions though can be challenging and a systematic approach to identify and sort the factors is needed. For this, comparison of the factors between each bootstrap run and the initial PMF output, as well as with externally determined markers, is crucial. As a result, certain solutions that exhibit sub-optimal factor separation should be discarded. The retained solutions would then be used to test the robustness of the PMF output. Meanwhile, analysis of filter sample with the Aerodyne aerosol mass spectrometer and the application of PMF and bootstrap analysis on the bulk water soluble organic aerosol mass spectra has provided insight into the source identification and their uncertainties. Here, we investigated a full yearly cycle of the sources of organic aerosol (OA) at three sites in Estonia, Tallinn (urban), Tartu (suburban) and Kohtla-Järve (KJ, industrial). We identified six OA sources and an inorganic dust factor. The primary OA types included biomass burning, dominant in winter in Tartu accounting for 73%±21% of the total OA, primary biological OA which was abundant in Tartu and Tallinn in spring (21%±8% and 11%±5%, respectively) and two other primary OA types lower in mass. A sulphur containing OA was related to road dust and tire abrasion which exhibited a rather stable yearly cycle and an oil OA was connected to the oil shale industries in KJ prevailing at this site comprising 36%±14% of the total OA in spring. The secondary OA sources were separated based on their seasonal behaviour: a winter oxygenated OA dominated in winter (36%±14% for KJ, 25%±9% for Tallinn and 13%±5% for Tartu) and was correlated with benzoic and phthalic acid implying an anthropogenic origin. A summer oxygenated OA was the main source of OA in summer at all sites (26%±5% in KJ, 41%±7% in Tallinn and 35%±7% in Tartu) and exhibited high correlations with oxidation products of *a*-pinene like pinic acid and 3-methyl-1, 2, 3-butanetricarboxylic acid (MBTCA) suggesting a biogenic origin.

## 1. Introduction

Particulate matter of aerodynamic diameter smaller than 10 μm ($PM_{10}$) has been extensively explored in many sites around the globe due to their various adverse effects upon human health and climate. In Europe, several monitoring networks have been measuring $PM_{10}$ for long time periods and an increasing trend in concentrations from north to south was noticed (Fuzzi et al., 2015, Putaud et al., 2010). Despite of this, some North European



countries are still suffering from $PM_{10}$ daily limit exceedances (European Environment Agency report No13/2017), and according to modelling studies following the "current legislation" scenarios, some of these sites will remain exposed to high $PM_{10}$ standards up to 2030 (Kiesewetter et al., 2015). Therefore, understanding the origins of the pollutants can play a crucial role for future abatement policies.

While large efforts have been devoted to the investigation of the sources and the chemical composition of $PM_{10}$ and in particular the organic fraction in western and central Europe, measurements in Eastern Europe and the Baltic region are scarce. More specifically, organic aerosol (OA) composition in Estonia has received little attention so far. $PM_{2.5}$ organic aerosol source apportionment was extensively studied by Elser et al. (2016) who performed mobile lab measurements during March 2014 at two different sites, Tallinn and Tartu. They found

similar sources of OA at both sites where residential biomass burning, traffic and long-range transported OA were the major sources of OA. They also found a localised residential influenced OA factor, which was connected to cooking activities and possibly coal and waste burning. Whilst the long-range transported OA was dominating during night time and during several events when polluted air masses were transported from Northern Germany, the remaining factors were important during day time. These results provided insights into

the spatial resolution of OA; nevertheless, they were limited to short time periods, hence, the seasonal variation of the pollutants remains unknown. Residential wood combustion and traffic were also presented as important sources of PM in previous long-term air pollution studies in Estonia (Urb et al., 2005, Orru et al., 2010). However, they did not provide any quantitative source apportionment on OA.

The offline AMS technique was recently developed by Daellenbach et al. (2016) where aqueous filter extracts

are measured after nebulisation with an Aerodyne high resolution time of flight aerosol mass spectrometer (HR-ToF-AMS, Canagaratna et al., 2007) and the resulting organic mass spectra are analysed with positive matrix factorisation (PMF, Paatero 1997). This technique has significantly increased our capability in investigating and identifying the seasonal behaviour of OA sources at several sites around the globe (Huang et al., 2014, Daellenbach et al., 2017, Bozzetti et al., 2017a). In addition, this technique allows for OA measurements of

different size fractions overcoming the limitation given by the transmission window of the AMS, resulting in quantifying sources from the coarse mode, such as primary (i.e. OA directly emitted in the atmosphere) biological  (Bozzetti et al., 2016) or sulphur containing primary OA sources (Daellenbach et al., 2017).

PMF is widely used to analyse ambient aerosol measurement data by decomposing the input aerosol mass spectra into factor concentration time series and factor profiles. To do so, PMF iteratively solves the bilinear Eq.

(1), where $\mathbf{X}i,j$ represents the measured input data matrix in which $i$ is the number of samples and $j$ the chemical species measured, $\mathbf{G}i,k$ represents the concentration time series matrix in which $k$ is the number of factors, $\mathbf{F}k,j$ represents the factor profiles matrix and $\mathbf{E}$ the residual matrix. The goal of PMF is to solve Eq. (1) such that the object function $\mathbf{Q}$ (Eq. 2) is minimised. In Eq. (2), $\mathbf{U}$ represents the corresponding error matrix.

$$X_{i,j} = \sum_k G_{i,k} F_{k,j} + E_{i,j} \qquad\qquad (1)$$


$$Q = \sum_i \sum_j [E/U]^2 \qquad\qquad (2)$$

Bilinear models suffer from rotational ambiguity; that is mathematically similar goodness of fit (Henry, 1987), leading to uncertainties in extracting the contributions of different OA sources. Additional modelling errors may occur due to the user subjectivity in analysing natural phenomena, when for example selecting the number of

interpretable factors or estimating the error matrix.

The bootstrap analysis (Davison and Hinkley, 1997), a resampling technique of the original data and error matrices, has been widely adopted to assess to a certain extent the rotational ambiguity related to PMF analysis (Brown et al., 2015). For each bootstrap iteration a random number of samples are selected with repeats from the original input matrices (*base case*), to recreate new input matrices with the same dimensions (*bootstrap*

*iteration*) that will be analysed with PMF. As a result, the bootstrap analysis results in a great number of solutions, whose environmental representativeness has to be assessed, which requires a systematic approach in





relating the separated factors to specific sources. This factor classification has been typically based on the contributions of certain markers (e.g. $C_2H_4O_2^+$ to identify biomass burning OA or $CO_2^+$ to identify secondary OA) (Daellenbach et al., 2017, 2018, Bozzetti et al., 2017a, Vlachou et al., 2018) in the case of the application of PMF to AMS data. However, in datasets with similar factor profiles, for example two oxygenated OA factors

or more, this type of sorting can become challenging, especially when a bootstrap iteration does not provide the expected factors. To assess the quality of the different bootstrap solutions, users typically compare the factor time-series to external marker measurements, when available, and discard suboptimal solutions using a set of acceptance criteria (Ulbrich et al., 2009, Zhang et al., 2011, Norris and Brown, 2014, Bozzetti et al., 2017a, 2017b, Zhang et al., 2017).

In this study, we propose a novel technique to evaluate PMF solutions generated through a large number of bootstrap iterations. The method is based on the examination of the correlation matrix between the base case and bootstrap iterations for both time series and profiles, without setting a priori a defined threshold in the correlation coefficient. We assess the performance of the technique (accuracy and probability of false positive and false negative results) by comparing the factors' time-series to the available specific markers. We have

applied the technique to an unprecedented dataset of 150 $PM_{10}$ filter samples from 3 sites in Estonia covering a full year (Sept. 2013- Sept. 2014), where anthropogenic and natural emissions of primary and secondary organic aerosols could be extracted.

## 2. Methods

### 2.1 Sampling sites

The samples were collected at three different sites in Estonia: Tallinn, Tartu and Kohtla-Järve. Tallinn is the capital and the largest city of Estonia located on the northern coast facing the Gulf of Finland. The measurement station is located about 9 km from the city centre, in the sub-district Õismäe (59°24′50.927″, 24°38′57.207″, 8.5 m a.s.l.). Tartu is the second largest city of Estonia, located in the south-eastern part of the country, in the valley of the Emajõgi River, a location that favours temperature inversions and the trapping of air pollutants. The

measurement station (58°22′14.183″, 26°44′5.517″, 39.5 m a.s.l.) is positioned in the city centre. According to Orru et al. (2010) traffic and local heating are important sources of air pollution at these sites. In both cities the fleet of cars increases in contrast to the limited street network capacity. Moreover, the local heating is more pronounced in Tartu compared to Tallinn. Kohtla-Järve (KJ) is a coastal industrial city located in the north-eastern part of Estonia (59°24′34.513″, 27°16′43.166″, 55.5 m a.s.l.). The main industries are related to large

production of petroleum products, oil shale processing and electricity generation. As it is not a highly populated area, residential heating or traffic are not as important as in the other two cities.

The measurements were performed with 24 h integrated $PM_{10}$ quartz fibre filter samples from KJ (31/08/2013 to 25/08/2014), Tallinn (05/09/2013 to 01/09/2014) and Tartu (05/09/2013 to 31/08/2014). PM was collected onto 15 cm diameter quartz filters, using a high volume sampler (500 l min$^{-1}$). After exposure, the filter samples were

wrapped in lint-free paper, sealed in polyethylene bags and stored at -20°C.

### 2.2 Major ionic species, sugar and acid analyses

The additional filter measurements, performed to corroborate and support the source apportionment, are listed in Table 1.

### 2.3 Offline AMS technique

The offline AMS technique was established by Daellenbach et al. (2016) and is briefly described in the following. From each filter sample, 4 punches of 16 mm diameter were extracted in 15 ml of ultrapure water (18.2 MΩ cm at 25 °C with total organic carbon < 3 ppb). The liquid extracts were inserted into an ultra-sonic bath for 20 minutes at 30°C. The ultra-sonicated samples were then filtered through a nylon membrane syringe of 0.45 µm. Out of the resulting solutions, aerosols were generated in Ar (≥ 99.998% Vol., Carbagas, 3073,

Gümligen, Switzerland) via an Apex Q nebulizer (Elemental Scientific Inc., Omaha, NE, USA) operating at 60°C and subsequently directed into the AMS after getting dried by a Nafion dryer.



The technique was performed on 150 filter samples in total: 39 from KJ, 69 from Tallinn and 42 from Tartu. The resulting organic spectra were analysed by PMF with the use of the Multilinear engine 2 (ME-2; Paatero, 1999). The interface for the data processing was provided by the Source finder toolkit (SoFi version 4.9; Canonaco et al., 2013) for Igor Pro (Wavemetrics Inc, Portland, Oregon, USA).

The $NH_4NO_3$ artefact on the $CO_2^+$ signal (Pieber et al., 2016) was also accounted for. For a thorough description of the artefact and the correction procedure the reader is referred to Pieber et al. (2016) and to Daellenbach et al. (2017).

**2.4 PMF input and uncertainties**

As stated in the introduction, the input data matrix for the PMF is the organic mass spectra data matrix $X_{i,j}$ that
consists of a combination of factor profiles, and time-series and the input error matrix **U** includes the blank variability and the measurement uncertainties. Before using the PMF algorithm, all the fragments with signal-to-noise ratio (SNR) below 0.2 were removed from the input matrix and the ones with SNR below 2 were down-weighted according to Paatero and Hopke's (2003) recommendations. Note that the PMF input matrix $X_{i,j}$ included the data from all 3 sites.

To obtain quantitative results, both data and error matrices were multiplied by the externally measured water soluble OC (WSOC) times the OM:OC ratio retrieved from the factor profiles of the matrix $X_{i,j}$.

Even though traffic is expected to be one of the primary sources of air pollutants especially in Tallinn and Tartu, a clear hydrocarbon like OA (HOA) which mainly contains non water soluble compounds could not be identified due the extraction procedure used. To assess a potential effect of the water soluble HOA (WSHOA)
on the PMF results we estimate the WSHOA contribution to the different fragments in the data matrix. The calculation was based on the time series of the concentration of EC, and the averaged high resolution HOA reference factor profiles from Crippa et al. (2013) and Mohr et al. (2012) multiplied by the HOA/EC ratio (=0.4) reported by El Haddad et al. (2013) times the recovery $R_{HOA} = 0.11$ reported by Daellenbach et al. (2016) (see Section 4 for more information on the recoveries). The calculated WSHOA data matrix was then subtracted
from the original data matrix. The PMF output did not change with the subtraction of WSHOA even though the calculated concentration of the latter was a high estimate due to the assumption that EC only originates from traffic. A thorough apportionment of EC and the calculation of HOC will be discussed in Section 4.1.

The variability of the AMS input dataset was best described by a seven factor solution which will be thoroughly described in Section 3.1 below. To assess the stability of the PMF solution and the sensitivity of the model for
the WSHOA subtraction, we performed 250 bootstrap runs by perturbing the HOA/EC and $R_{HOA}$ parameters within their errors (1 standard deviation, σ) assuming a normal distribution. Note that this number of runs was restricted by computational limitations.

The sorting of the factors and the concomitant selection of the retainable solutions out of the 250 runs was based on the correlation ($R$ Spearman, $R$s) of the time series between the base case (which is the PMF result before
bootstrapping) and each bootstrap iteration $n$, as well as the correlation ($R$s) of each factor profile between the base case and each iteration $n$. In the following, the sorting based on the time series is denoted as "ts" and the sorting based on profiles is denoted as "pr". The criteria, followed for the selection or rejection of each solution, are described in Section 3.2.

**3. Source apportionment**

**3.1 Interpretation of PMF factors**

As already mentioned, the variability of the water soluble organic mass spectra was best explained by a seven factor solution, to which we refer as *base case*. The factors found were:

    1.  An oil related OA which was rich in hydrocarbons (Fig. 1a) and showed elevated concentrations mainly in KJ (Fig. 1b).





2. A sulphur containing OA (SCOA) factor with a pronounced peak at $m/z$ 79 ($CH_3SO_2^+$) (Fig. 1a) and rather stable contributions at all sites (Fig. 1b). As this factor was significantly enriched in the coarse mode (Vlachou et al., 2018), mainly found in urban areas influenced by traffic at other European sites (Daellenbach et al., 2017) and clearly associated with fossil carbon (Vlachou et al., 2018), we have previously related it to asphalt abrasion or tire wear.

3. A summer oxygenated OA (SOOA) with enhanced $m/z$ 43 ($C_3H_2O^+$) and 44 ($CO_2^+$) peaks (Fig. 1a) which was highest during summer at all sites (Fig. 1b).

4. A winter oxygenated OA (WOOA) with enhanced peaks at $m/z$ 28 ($CO^+$) and 44 ($CO_2^+$) (Fig. 1a) dominating in winter at all sites (Fig. 1b). Both of these two oxygenated factors (SOOA and WOOA) were also found in different European sites and were connected to non-fossil sources; biogenic and anthropogenic respectively (Vlachou et al., 2018, Daellenbach et al., 2018). Such a distinction was also found in Canonaco et al. (2015) where an online ACSM was used.

5. A factor with a significantly pronounced peak at $m/z$ 44 ($CO_2^+$) (Fig. 1a) and elevated concentrations in summer (Fig. 1b), which was identified as dust. This factor will be more thoroughly examined in Section 4.

6. A primary biological OA (PBOA) which exhibited high contributions of the fragment $C_2H_5O_2^+$ (at $m/z$ 61, Bozzetti et al., 2016) (Fig. 1a) and increased concentrations during late spring and summer at all sites (Fig. 1b).

7. A biomass burning OA (BBOA) with a characteristic peak at $m/z$ 60 ($C_2H_4O_2^+$) (Alfarra et al., 2007) (Fig. 1a) and elevated concentrations during late fall and winter in Tallinn and especially Tartu (Fig. 1b).

**3.2 PMF uncertainty analysis: factor sorting and solution selection**

The framework for factor sorting and solution selection proceeded as follows:

1. A correlation matrix was composed including all the correlations between base case factor time series (profiles) represented in rows and bootstrap iteration $n$ time series (profiles) represented in columns, demonstrating the $R$s per correlation (Fig. 2).

2. Factors were sorted according to the highest correlation of their time series (profiles) with the base case factor time series (profile).

3. Solutions were discarded, if any of the correlation coefficients occurring in the matrix diagonal was not statistically significantly higher than at least one of the coefficients in the respective column or row (significance level α=0.05). These selection criteria have two implications. (1) Every factor separated in a bootstrap run should correspond to a unique factor of the seven factors separated in the base case. (2) All factors that could be identified in the base case have one unambiguous corresponding factor in the bootstrap run. We have statistically evaluated the comparison between the Spearman coefficients by treating them as if they were Pearson coefficients (Myers et al., 2006) and applying the standard Fisher's z-transformation and subsequent comparison using a $t$-test. This approach was reported to be more robust with respect to Type I error (false positive) than ignoring the non-normality and using Pearson instead of Spearman coefficients.

An example of the correlation matrix of a retained solution is shown in Fig. 2a for time series and 2b for profiles. Meanwhile, Fig. 2c and 2d represent an example of a bootstrap iteration ($n$=140) where the solution was rejected because SCOA was not resolved, based on both ts and pr analysis, respectively. In Fig. 2c, the highest correlation between the time series of factor 2 and the factors of the base case was found with Dust instead of SCOA, yet much weaker ($R$s=0.34) than the correlation between factor 7 and Dust ($R$s=0.79). Therefore, factor 7 could be identified as Dust and factor 2 could not be identified as an interpretable factor. In Fig. 2d factor profile 2 correlated most with the base case SCOA profile; however this correlation was not significantly higher than the correlation between factor 2 and Dust. Moreover, base case SCOA correlated better with factor 6, which was related to SOOA, indicating that SCOA could not be unambiguously related to factor 2.



To validate the selection of the solutions we compared the factors of each bootstrap iteration with an externally measured marker; more specifically BBOA with levoglucosan, PBOA with cellulose, WOOA with phthalic acid and SOOA with 3-methyl-1, 2, 3-butanetricarboxylic acid (MBTCA). The retained solutions exhibited the highest correlations between the external markers and the respective factors (red markers in Fig. 3). To seal the

validity of the retained solutions, we also compared the $R$s between the base case factors and their respective external marker with the $R$s between bootstrap iteration factors and their respective external marker. We performed the bootstrap technique for a second time on the time series of the base case factors and the respective external markers for 1000 times. The resulting $R$s coefficients are represented in probability density functions (PDF) indicated as red curves in Fig. 3, centred at 0.8 for BBOA (Fig. 3a), 0.7 for WOOA (Fig. 3c)

and 0.9 for SOOA (Fig. 3d) and a broader one centred at 0.45 for PBOA (Fig. 3b). In all four cases the retained solutions, either coming from the "ts" or the "pr" approach, spanned around the centre of each PDF (Fig. 3 and S1 for the "pr") and most of the solutions where at least one factor was not resolved (black markers in Fig. 3) were not included within the PDF boundaries. The general agreement between the PDFs and the retained solutions ratified the solution selection approach, however, there were still some cases of possible misallocation

of retained or rejected solutions (for example a few black markers appearing at the centre of the PDF, Fig. 3c).

To assess whether the retained bootstrap solutions share the same quality with the base case solution with regard to correlations between factors and markers, we calculated the probability of type I (false positive) and type II (false negative) errors associated with the solution selection approach (Fig. 4). The analysis entailed a quantitative comparison of the Spearman coefficients obtained between markers and factor time series from the

bootstrap iterations, $R$s$_{boots}$, with the respective Spearman coefficients obtained between markers and factor time series from the base case $R$s$_{base}$, considering the same samples as in the corresponding bootstrap iterations. The comparison between $R$s$_{boots}$ and $R$s$_{base}$ was performed by applying a Fisher transformation followed by a $t$-test. We defined true positive and false negative as the red (retained solutions) and black (non-resolved factors) points respectively lying within the PDF boundaries with regard to the total number of red and black points

within the PDF boundaries. True negative and false positive were defined as the black and red points lying outside the PDF boundaries with respect to the total number of red and black points outside the PDF boundaries. The limits between false positive and false negative were set by 2 standard deviations from the one-to-one line. The percentages of the accuracy and the probability of false positive or false negative cases are compiled in Table S1. Sorting based on profiles seemed less reliable and is more prone to false negative solutions (Table S1

and S2), as the profiles often look similar and therefore the $R$s exhibits high values for all factors (Fig. 2b and d). On the contrary, sorting based on time series showed clearer results as the $R$s spanned over a greater range of values (Fig. 2a and 2c). Still the "ts" method produced false negative solutions, for example 53% for PBOA due to the combination of (i) limited number of points available for cellulose and (ii) the representativeness of the marker time series after the resampling (bootstrap analysis). Note that PBOA was important only during a few

days in spring and therefore it is possible that these days were not always selected in the resampling process. The SOOA on the other hand exhibited 0% false negative and 16% false positive cases always demonstrating high $R$s$_{boots}$ and $R$s$_{base}$ values.

However, in any of the two methods "ts" and "pr", the Fisher-transformed correlation coefficient rendered the selection of the solution evident, and eventually the two sorting methods yielded a very similar retained solution

space (Fig. 4 and 5). Figure 5 depicts the correlation between the averaged common retained solutions and the averaged retained solutions coming from either the "ts" or the "pr" sorting method for the example of BBOA (correlations for the other factors are shown in Fig. S2). There is a minor deviation from the one to one line for the standard deviation scatter plot (Fig. 5b) for the "ts" sorting method. However, as soon as the solutions got weighted according to the correlation between external marker and bootstrap run time series, then the deviation

decreased (blue markers in Fig. 5). The weighting factor $w_i$ was calculated as:

$$w_i = \frac{1}{\sqrt{\sum_i (SE)^2}}$$    (4), where SE is the standard error resulting from the regression between external marker and bootstrap iteration.



## 4. Investigation of sources and discussion

### 4.1 Estimation of traffic contribution to EC and OC

We estimated above that the traffic contribution to WSOA (< 1%) can be neglected and has little effect on the PMF outputs. However, traffic might be an important source of EC and OA, which is assessed in the following.

To estimate the percentage of EC coming from traffic ($EC_{tr}$) we used the ME-2 model (with SoFi standard version 6.399, Canonaco et al., 2013) assuming that the sources of EC are biomass burning, resuspension of road dust, industrial emissions from the oil shale factories and traffic. Here the input data matrix included the time series of water soluble BBOA (WSBBOA), WSSCOA, WSOilOA and EC. PMF was called 1000 times varying the initial seed to solve Eq. (3).

$$EC = EC_{tr} + EC_{bb} + EC_{oil} + EC_{rrd} = EC_{tr} + a*WSOA_{bb} + b*WSOA_{oil} + c*WSOA_{soil} \qquad (3)$$

Here, $EC_{bb}$, $EC_{oil}$ and $EC_{rrd}$ represent the EC concentration time-series related to biomass burning, oil shale industry and resuspension of road dust, respectively, and a, b, and c are the EC/WSOA ratios characteristic of the emissions from the same sources. This new methodology, based on PMF, is especially pertinent as unlike other inversion techniques it sets positive constraints on a, b and c and offers the possibility of resolving the
contribution of factors for which no constraints are available, here $EC_{tr}$.

We found that $EC_{tr}$ contributed 57% to the total EC (on a yearly average), while 36% of EC was attributed to biomass burning, 2% to road dust resuspension and 4% to the oil shale emissions. The contribution of EC from fossil fuel combustion ($EC_{ff}$) measured in a similar site to Tartu, i.e. an Alpine valley in Southern Switzerland, Magadino in 2014 (Vlachou et al., 2018) was in agreement with our $EC_{tr}$ contribution as the yearly average was
55%±7%. Also in Zurich, an urban site, $EC_{ff}$ ranged from 40% to 55% during winter 2012 (Zotter et al., 2014). From the $EC_{tr}$ contribution, we estimated that the HOC (multiplication of the $EC_{tr}$ time series with the ratio HOC/EC) contributed 4% to the total OC on a yearly average.

### 4.2 Scaling to organic carbon

All the retained solutions (in total 62%) were averaged per factor and their seasonal behaviour as well as their
correlations with the external markers are presented in Section 4.4. Note that all the water soluble factors were recovered following the method described in Daellenbach et al. (2016) and Vlachou et al. (2018). The recoveries were calculated by fitting Eq (4).

$$OC_{i,n} = \sum_k \frac{WSOC_{i,n,k}}{R_k} \qquad (4)$$

where $OC_{i,n}$ is the OC concentration per bootstrap run $n$ per sample $i$, $R_k$ is the recovery $R$ per factor $k$, and
$WSOC_{i,n,k}$ is the water soluble OC concentration calculated based on the measured WSOC and the OM:OC per factor. From the $OC_{i,n}$ the part of hydrocarbon like OC as well as the inorganic carbon related to dust were removed. The carbonate carbon investigation and calculation is discussed in Section 4.3. To define the recoveries we used a non-negative multilinear fit. The starting points of the fitting for each $R_k$ with the exception of $R_{oil}$ were obtained from the literature (Bozzetti et al., 2016, Daellenbach et al. 2016 and Vlachou et
al., 2018) and were randomly varied within their literature range with an increment of $10^{-4}$. The final distributions of the recoveries are shown in the Supplementary (Fig. S3). The recoveries in this study were all shifted to the lower end of the recoveries reported in the literature. While the reason remains unclear, the water solubility of OA is dataset specific therefore we can expect differences to other datasets. Moreover, we re-measured in a different laboratory the OC concentrations from a subset of 21 samples covering all sites and all
seasons. The agreement between the two differently determined OC concentrations was excellent (Fig. S4, slope = 0.93, $R^2 = 0.99$).

### 4.3 Exploration of the dust factor

Mineral dust can contain a significant amount of inorganic carbon in the form of $CO_3^{2-}$. The water extracts used in the offline AMS technique have a pH that is always < 8. Therefore, the $CO_3^{2-}$ in our samples is all solubilized
into $HCO_3^-$ (pKa ($HCO_3^-/CO_3^{2-}$) = 10.33, Bruice, 2010).The latter is shown to release $CO_2$ when it undergoes



thermal decomposition on the AMS vaporiser (at 600°C) (Bozzetti et al., 2017b). Thus, the contribution of $CO_2^+$ to organic aerosols is overestimated and the fraction coming from the inorganic carbon should be removed from the OA spectra.

To remove the influence of the inorganic dust, we estimated the carbonate carbon concentrations ($C\_CO_3$) corrected for the relative ionisation efficiency, as discussed in the Supplementary. This estimated $C\_CO_3$ concentration was compared to measured carbonate on a subset of 19 filter samples. While the agreement between measured and estimated concentrations is poor for Tartu, a decent agreement had been found for KJ and Tallinn, especially given the large uncertainties in both variables (Fig. S5). On an absolute basis, PMF seems to overestimate the $C\_CO_3$ concentrations by ~20% compared to the measured concentrations.

$Ca^{2+}$ is one of the most common constituents of mineral dust and therefore can be used to trace this source. The time series of $C\_CO_3$ and $Ca^{2+}$, available for the entire set of samples, displayed in Fig. 6, show that the two variables exhibit similar trends (except for Tallinn). Despite the large errors in $C\_CO_3$ estimates, an uncertainty weighted correlation analysis (SI and Fig. S6) shows that $C\_CO_3$ and $Ca^{2+}$ correlate statistically significantly ($R$=0.4, p-value<$10^{-5}$) with a slope of 0.2, consistent with $C\_CO_3$ and $Ca^{2+}$ being in the form of calcium bicarbonate.

We have validated the identity of the dust factor even further, by measuring the same subset of 19 filters with the offline AMS technique before and after fumigation with HCl (as described in Zhang et al., 2016). The comparison of the mass spectra of fumigated and non-fumigated samples is illustrated in Figure 7 for two samples: with high and low contribution of dust. In the example of KJ (05/06/2014) where the dust factor exhibited the highest contribution, $f44$ was substantially decreased after fumigation (Fig. 7a). In the case of Tallinn (19/01/2014), where the dust factor concentration was negligible, $f44$ remained stable after fumigation (Fig. 7b). Overall, the comparison of the $\Delta f44$ modelled (= $f44total$ – $f44org$) from the initial data set of 150 filters and the $\Delta f44$ measured (=$f44non\_fum$ – $f44fum$) from the subset of 19 filters showed consistent results (Fig. 8). Taken together these results provide strong confidence on the nature of the dust factor extracted by PMF.

### 4.4 Seasonal variation of organic aerosol sources

The sources were quantified after removing the contribution of the dust factor from the total OA. All the factor concentrations with their uncertainties averaged per season are presented in Table S3. In general, the relative uncertainties decreased with increasing concentrations per factor (Fig. S7). For concentrations above ~1 µg m$^{-3}$ the percentage error became more important than the error related to noise and thus more stable for all factors. Consistent with the factor separation and uncertainty analysis above, the factors that were well separated, such as SOOA, exhibited low relative uncertainty (0.15), while the factors that were more difficult to extract, such as BBOA exhibited higher relative uncertainty (0.45).

BBOA exhibited high concentrations in Tallinn during winter (on average 3.7±2.7 µg m$^{-3}$) and fall (1.2±0.9 µg m$^{-3}$) and in Tartu (8.4±3.9 and 3.8±1.9 µg m$^{-3}$, winter and fall respectively) (Fig. 9a, Table S3). In both cities BBOA and the marker levoglucosan correlated very well (Fig. 9b) confirming the identity of the factor. For KJ the concentrations of BBOA were lower in winter (1.3±0.8 µg m$^{-3}$) as expected, due to the low number of residents and low biomass burning activities in the region. At all sites the levoglucosan:BBOA ratio (0.08 for KJ, 0.05 for Tallinn and 0.05 for Tartu) assessed in this study was consistent with the one reported in the neighbouring country, Lithuania (Bozzetti et al., 2017). Potassium ($K^+$), which is often used as a wood burning marker, especially for ash, also correlated well with BBOA for Tallinn ($R^2$=0.80) and Tartu ($R^2$=0.58). Different BBOA:$K^+$ ratios were used in the past to describe burning conditions (Zotter et al., 2014, Daellenbach et al., 2018) and higher values were linked to inefficient burning conditions. Here, the BBOA:$K^+$ ratio (14.3 for Tallinn and 18.1 for Tartu) was in agreement with the one found at Southern Alpine valley sites (Magadino and San Vittore, Switzerland, Daellenbach et al., 2018) where older stoves are still used. In Estonia more than 80% of households use old type stoves for heating purposes (Maasikmets et al., 2015), therefore, BBOA could be linked to inefficient residential wood burning.



The recognition of PBOA as described in Section 3.2 was also supported by the high correlations of this factor with cellulose and erythritol (Fig. 9d). Cellulose is related to plant debris and is typically used as a marker for primary biological aerosols (Bozzetti et al., 2016) while, erythritol among other sugar alcohols, reflects airborne biogenic detritus (Graham et al., 2002). The seasonal behaviour of PBOA was very similar to the respective

behaviour of both markers (Fig. 9c) with average spring concentrations of 0.2±0.2 µg m$^{-3}$ for KJ, 1.2±0.8 µg m$^{-3}$ for Tallinn and 0.7±0.4 µg m$^{-3}$ for Tartu.

SOOA exhibited a clear yearly cycle at all sites with the highest concentrations witnessed in summer and early fall (in summer on average 1.8±0.7 µg m$^{-3}$ for KJ, 2.8±0.6 µg m$^{-3}$ for Tallinn and 2.1±0.4 µg m$^{-3}$ for Tartu) (Fig. 9e). In previous studies this factor showed an exponential increase (Fig. S8) with temperature and was linked to

terpene oxidation products (Daellenbach et al. 2017, Leaitch et al., 2011). In another study in an Alpine valley site, this factor was also found to be 79% non-fossil, supporting the connection to biogenic secondary OA (Vlachou et al., 2018). Here, SOOA was not only correlating with temperature ($R$s = 0.77, Fig. S7) but also with two oxidation products of $a$-pinene, i.e., with pinic acid and even better with MBTCA (Fig. 9f). The latter was shown to be produced by reaction of pinonic acid with the OH-radical (Mueller et al., 2012).

WOOA was more pronounced during fall and winter at all sites with average concentrations in winter of 1.4±0.5 µg m$^{-3}$ for KJ, 2.2±0.8 µg m$^{-3}$ for Tallinn, and 1.5±0.5 µg m$^{-3}$ for Tartu (Fig. 9g). In earlier studies WOOA was characterised as non-fossil (Vlachou et al., 2018) and was identified based on its correlation with $NH_4^+$ coming mainly from $NH_4NO_3$ in winter time (Lanz et al., 2007, Daellenbach et al., 2017). It was also related to long-range transported oxygenated OA stemming from anthropogenic emissions during winter, such as biomass

burning. Also, WOOA demonstrated high correlations with two anthropogenic organic acids: benzoic and phthalic acid (Fig. 9h), formed via the photo-oxidation of aromatic hydrocarbons, such as toluene and naphthalene, and therefore suggested to be tracers for anthropogenic sources (Kawamura and Yasui 2005, Deshmukh et al., 2016). Recently, Bruns et al. (2016) found that aromatic compounds, such as benzene and naphthalene emitted by wood combustion can indeed produce highly oxidized SOA. Taking all the above into

account, it was concluded that WOOA might be linked mostly to aged wood burning OA.

Figure 10 illustrates the relative contributions per factor per site to the total OA (all averaged contributions per season per factor with their uncertainties are shown in Table S4). Out of the primary sources, the major contributor was BBOA during winter and fall (on average and one standard deviation: 39%±16% and 27%±13% in Tallinn and 73%±21% and 53%±14% in Tartu). However, in KJ during winter and fall WOOA was the

dominant source (36%±14% and 39%±13%, respectively) indicating that for this site regional transport of OA is important. In spring, PBOA was the major source in Tartu (21%±8%) while in Tallinn BBOA and SOOA were the dominating sources during that season (30%±14% and 18%±5%, respectively). This could be due to the fact that temperatures in early spring are still low (2°C on average in Tallinn in March) and wood burning for residential heating is still widely used. Towards the end of spring (15°C on average in Tallinn in May) the rising

temperature favours the biogenic emissions. In KJ, the most important source was oil OA (36%±14% in spring), most possibly coming from the oil shale industries in the region. The presence of the oil factor at the other two sites could be an indication that this factor is mixed with coal or waste burning, as also found by Elser et al. (2016). Besides, the oil OA profile resembled the coal profile identified in Cork city, Ireland (Dall'Osto et al., 2013). During the summer months and early fall, SOOA was prevailing over all sources at all sites, with

26%±5% in KJ, 41%±7% in Tallinn and 35%±7% in Tartu. Even though KJ is highly industrialised, SOOA can still be related to the production of secondary OA from biogenic volatile organic compounds. The least significant source, especially in Tartu, with rather stable seasonal behaviour was SCOA. The yearly average contribution of SCOA was 12%±4% in KJ, 14%±5% in Tallinn and 4%±2% in Tartu. In general the primary sources seem to dominate the secondary ones which is also observed in other European sites such as Payerne

(Bozzetti et al., 2016) or Magadino especially in winter (Vlachou et al., 2018) as well as in China, Beijing (Zhang et al., 2017).



## 5. Conclusions

The offline AMS technique was applied on a set of 150 filter samples covering a yearly cycle at three sites in Estonia. The uncertainties of the PMF solution were assessed by bootstrap analysis. In order to identify the factors the Spearman $R$ ($R$s) coefficients between base case time series and bootstrap run time series ("ts") as

well as base case profiles and bootstrap run profiles ("pr") were monitored. The results showed that the retained solution space if one follows the "ts" or the "pr" sorting method was very similar. Weighting with the $R$s between external markers and bootstrap run increased our confidence towards the solution space. The source apportionment results revealed four primary OA sources, two secondary OA and a dust factor. The dust factor was identified by measurements of calcium carbonate as well as by acidification with HCl of a selected batch of

filters. Out of the primary sources, three had an anthropogenic influence. BBOA was mainly present in winter and autumn in Tallinn and Tartu, the two largest cities of Estonia, where residential heating activities are common. SCOA was mostly important in winter in Tallinn and KJ in contrast to Tartu. The third anthropogenic primary factor was oil OA which exhibited the highest concentrations in KJ, as expected. The reason why this factor was evident in Tallinn and Tartu could be that it may include coal combustion for residential heating

purposes. PBOA was the only primary OA not related to anthropogenic emissions and was prevailing in spring at all sites. The two oxygenated OA factors were separated according to their seasonal behaviour: WOOA was linked to anthropogenic wood burning activities as it dominated in winter and autumn at all sites and also correlated with phthalic and benzoic acid. SOOA was significant during summer at all sites and was related to biogenic emissions and strong aging as it was highly correlated with a second generation oxidation product of $a$-

pinene.

**Competing interests.** The authors declare that they have no conflict of interest.

**Author Contribution.** AV performed the offline AMS measurements, data curation and formal analysis, AV and IEH wrote the paper, AT assisted with the formal analysis, HL provided expertise on AMS measurements,

FC and KD provided expertise on software, JLJ performed additional measurements including major ions, sugars, organic acids, cellulose, OC/EC and WSOC, MCM performed the carbonate measurements, MM and ET organised the filter sampling, UB, ASHP and IEH were involved with the supervision and conceptualisation. All authors commented on the paper and assisted in the interpretation of the results.

### Acknowledgements

This work was funded by the Estonian–Swiss cooperation program "Enforcement of the surveillance network of the Estonian air quality: Determination of origin of fine particles in Estonia". María Cruz Minguillón acknowledges the Ramón y Cajal Fellowship awarded by the Spanish Ministry of Economy, Industry and Competitiveness. The Labex OSUG@2020 (ANR-10-LABX-56) provided the funding for part of the analytical

equipment at IGE (France). We also acknowledge the contribution of the COST Action CA16109 COLOSSAL.

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

Table 1. Detailed overview of the additional measurements.

| Method | Compounds | Number of filters |
|---|---|---|
| Ion chromatography (Jaffrezo et al., 1998) | $K^+$, $Na^+$, $Mg^{2+}$, $Ca^{2+}$, $NH_4^+$, $Cl^-$, $NO_3^-$, $SO_4^{2-}$ and methane sulfonic acid | All (150) |
| Liquid chromatography-electrospray ionisation mass spectrometry (Jacob et al., 2018) | Organic acids (e.g. benzoic acid, pinic acid) | 69 (Tallinn) |
| High performance liquid chromatography with pulsed amperometric detection (Waked et al., 2014) | Anhydrous sugars (e.g. levoglucosan, mannosan) and sugar alcohols (e.g. erythritol, mannitol) | 150 |
| Enzymatic conversion of cellulose (Kunit and Puxbaum, 1996) | Cellulose | 69 (Tallinn) |
| Sunset EC/OC analyser (Birch and Carry, 1996) with the EUSAAR2 protocol (Cavalli et al., 2010) | Organic (OC) and elemental (EC) carbon | 150 |
| Total organic carbon analyser (Piot et al., 2012) with the use of catalytic oxidation and detection of $CO_2$ with a non-dispersive infrared detector | Water soluble OC (WSOC) | 150 |
| Thermal Optical Transmittance using Sunset EC/OC analyser (Karanasiou et al., 2011) | $CO_3^{2-}$ | 19 (from all sites) |



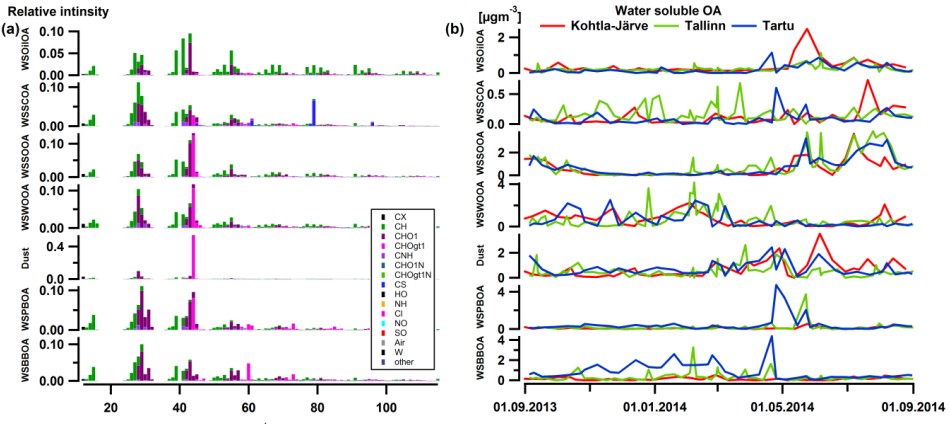

Figure 1. Factor profiles (a) and time series (b) of the seven factor solution: Biomass burning OA (BBOA), sulfur-containing OA (SCOA), primary biological OA (PBOA), Industrial/ Oil combustion OA (OilOA), winter oxygenated OA (WOOA), summer oxygenated OA (SOOA) and a dust related OA (Dust). Note that the water soluble parts are illustrated here.

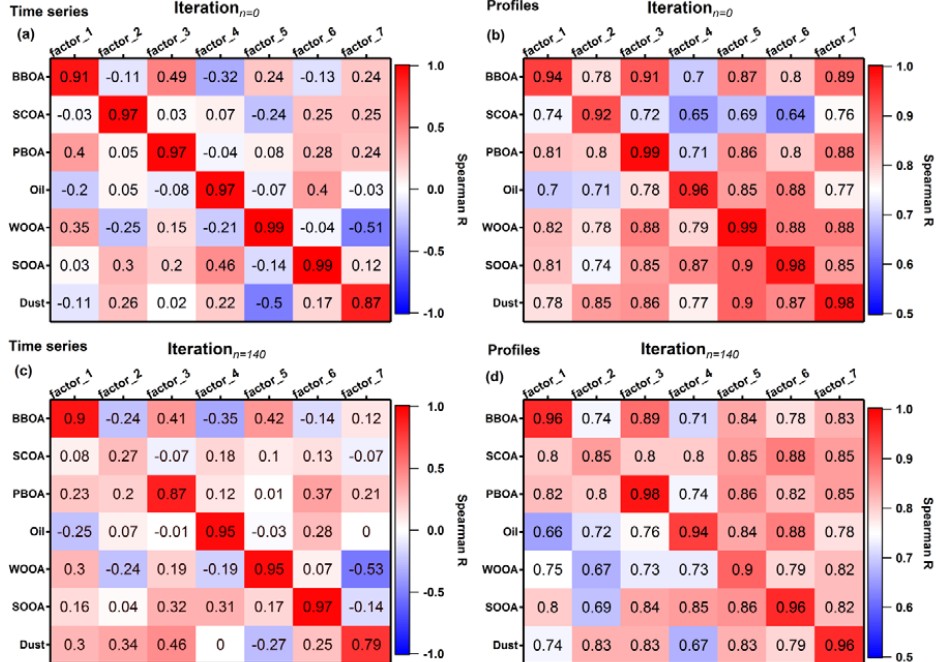

Figure 2. Explanatory tables for the factor sorting based on the time series: (a) accepted bootstrap iteration where all highest correlations ($R$s) lay in the diagonal, (c) failure to resolve SCOA as for this bootstrap iteration both factors 2 and 7 showed the highest correlation with Dust. The respective tables for the case of profiles are in (b) and (d). Note the different scales of $R$s for the profiles.



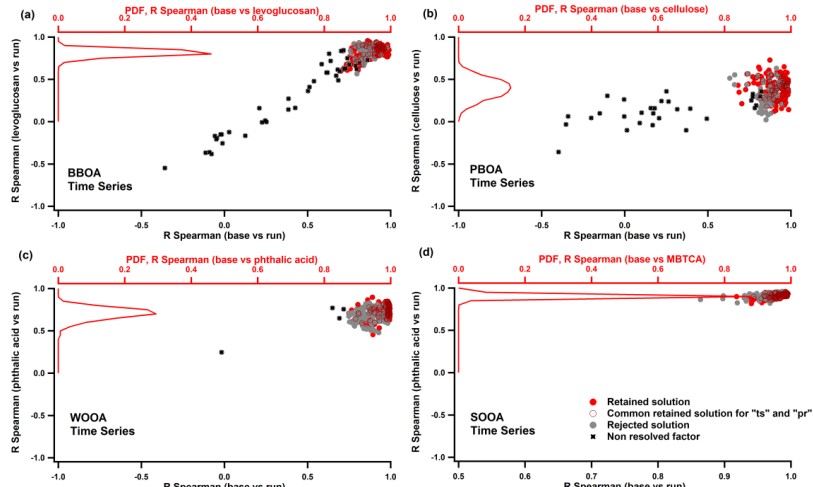

Figure 3. Solution space per factor defined by investigation of the correlation (*R*s) between base case time series and bootstrap runs (bottom x-axis) and external markers and bootstrap runs (y-axis): BBOA with levoglucosan (a), PBOA with cellulose (b), WOOA with phthalic acid (c) and SOOA with MBTCA (d). The retained solutions are indicated in red and the rejected ones in grey. The points in black represent the runs where the specific factor was not resolved at all. Each PDF (top x-axis) includes the range of *R* coming from the correlations between the time series of the base case factors with their respective markers.





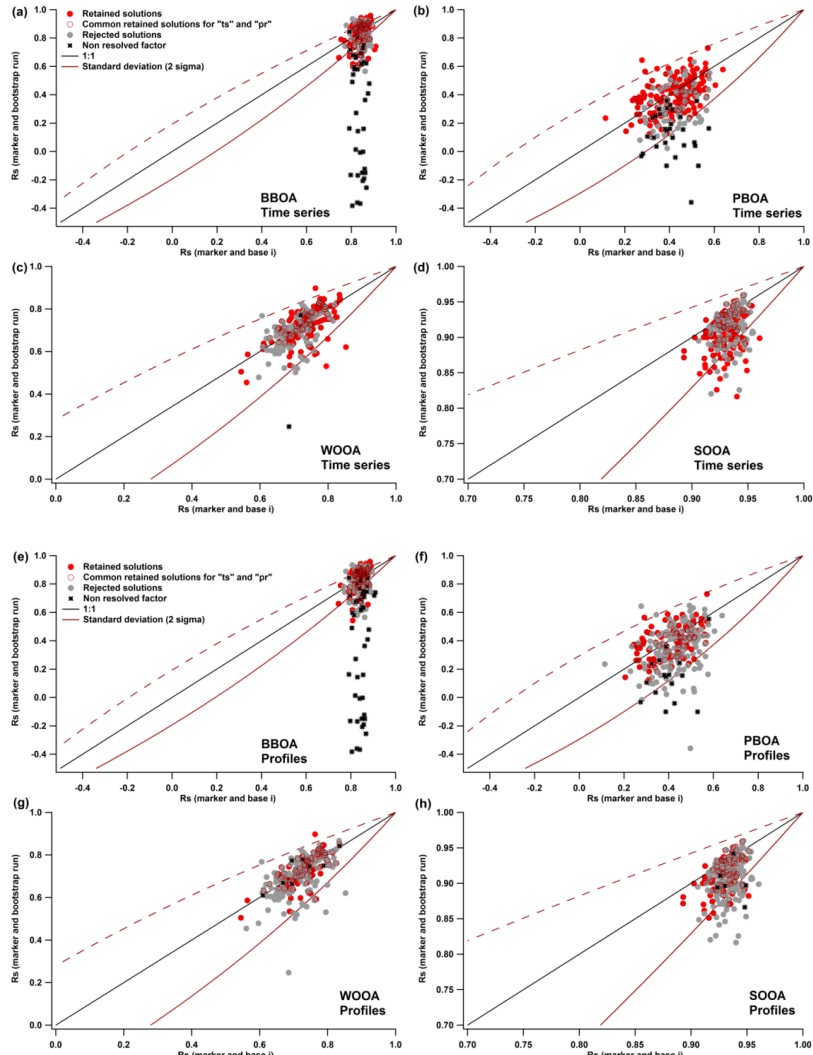

Figure 4. Scatter plots of $Rs_{boots}$ between markers and bootstrap runs and $Rs_{base}$ between markers and base cases per factor: BBOA with levoglucosan (a), PBOA with cellulose (b), WOOA with phthalic acid (c) and SOOA with MBTCA (d) for the "ts" sorting method. The black points that lie below the 1σ line (in solid dark red) indicate true negative solutions, whereas the black points within 2σ indicate false negative solutions. The red points below 1σ represent the false positive solutions, while the red points above are the true positive solutions. The respective scatter plots for the "pr" method are shown in (e), (f), (g) and (h). Note the different $Rs$ scales per factor.



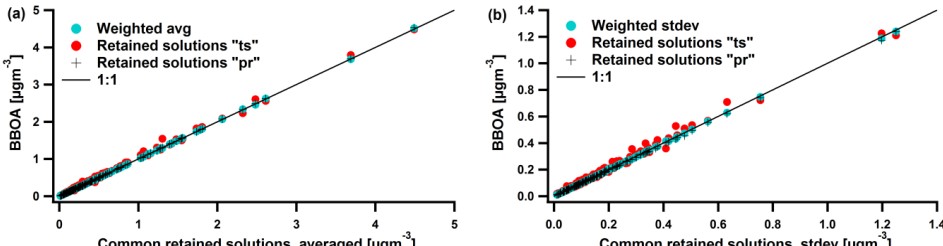

Figure 5. (a) Scatter plots between BBOA time series averaged (avg) over the common solutions coming from both the time series ("ts") and profile ("pr") sorting method plotted in the x-axis, and plotted in the y-axis the BBOA time series averaged over the solutions coming from the "ts" method in red, from the "pr" in black cross and from the weighted average based on the marker, here levoglucosan, in blue. The respective scatter plot for the standard deviation (stdev) is shown in (b).

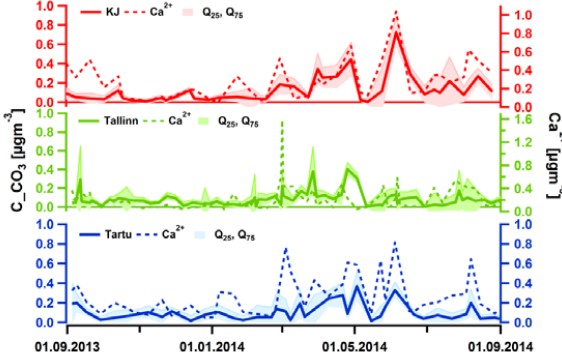

Figure 6. Time series of the inorganic dust factor (C_CO$_3$) with interquartile ranges (Q$_{25}$ and Q$_{75}$) and of Ca$^{2+}$ per site.

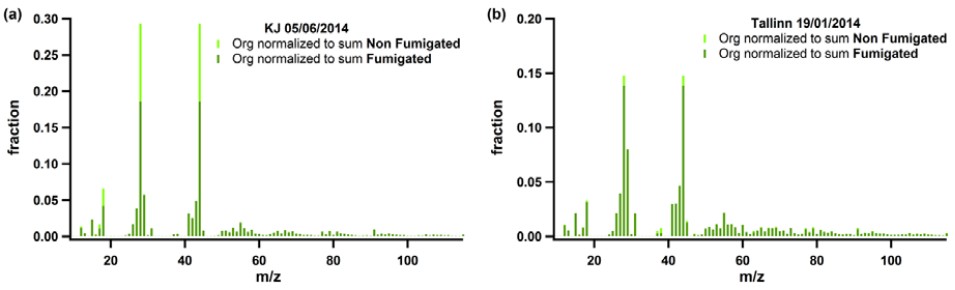

Figure 7. Examples of fumigated and non-fumigated mass spectra from two samples, with a high (KJ 05/06/2014, a), and a low dust concentration (Tallinn 19/01/2014, b).





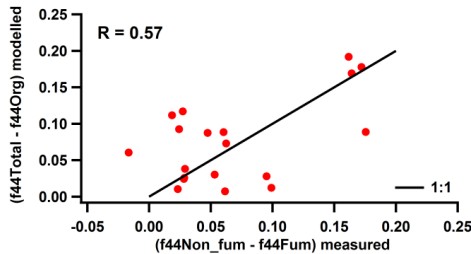

Figure 8. Scatter plot between the calculated Δ$f$44 and the measured Δ$f$44 coming from dust.



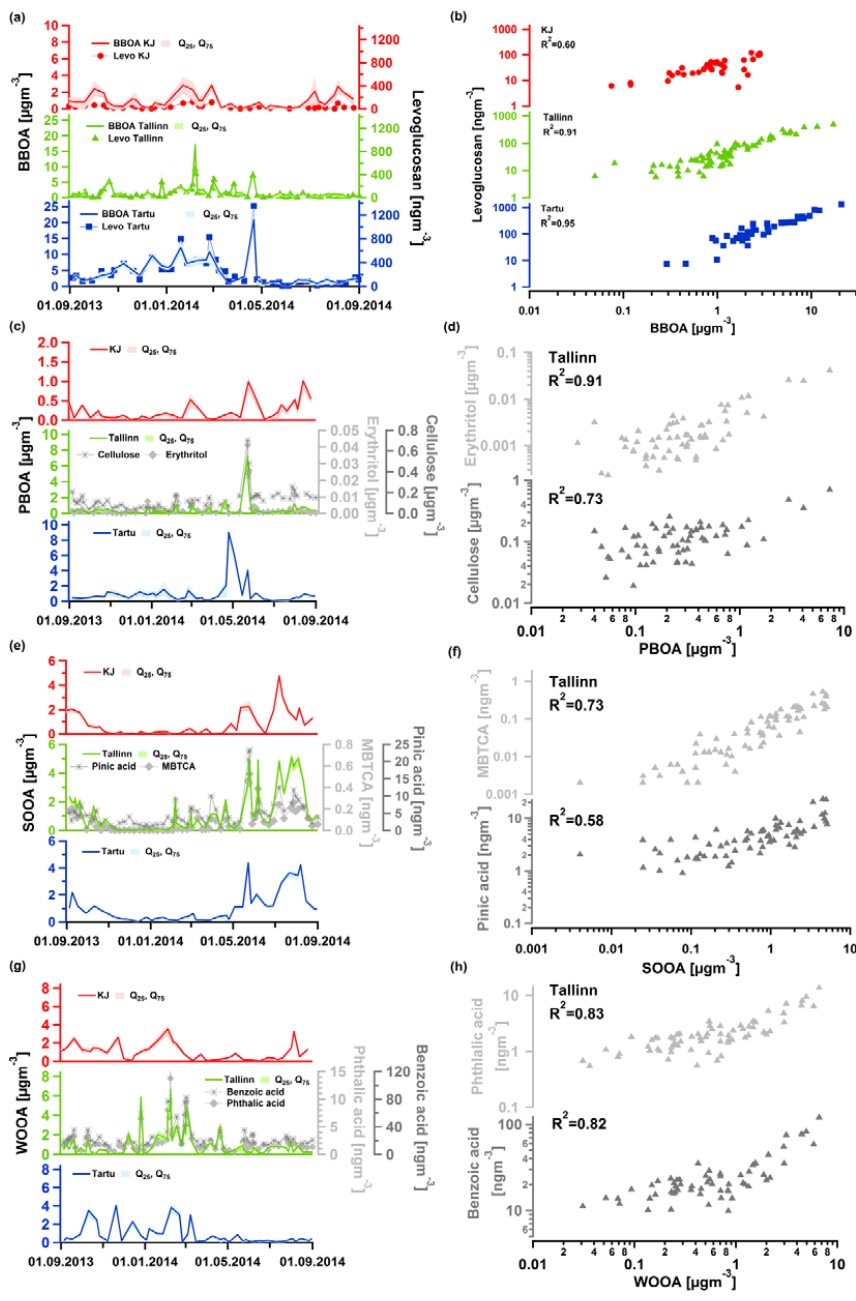

Figure 9. Factor median and external marker concentration time series for all three sites; (a) BOOA and levoglucosan (note the change of the scale for KJ), (c) PBOA, cellulose and erythritol, (note the change of the scale for KJ) (e) SOOA, pinic acid and MBTCA and (g) WOOA, benzoic and phthalic acid, with the respective scatter plots between factor and external marker in (b), (d), (f) and (h). The colours red, green and blue denote the site (Kohtla Järve, Tallinn and Tartu) and the markers in light and dark grey denote the concentrations of the external markers. The shaded areas represent the first ($Q_{25}$) and third ($Q_{75}$) quartiles.





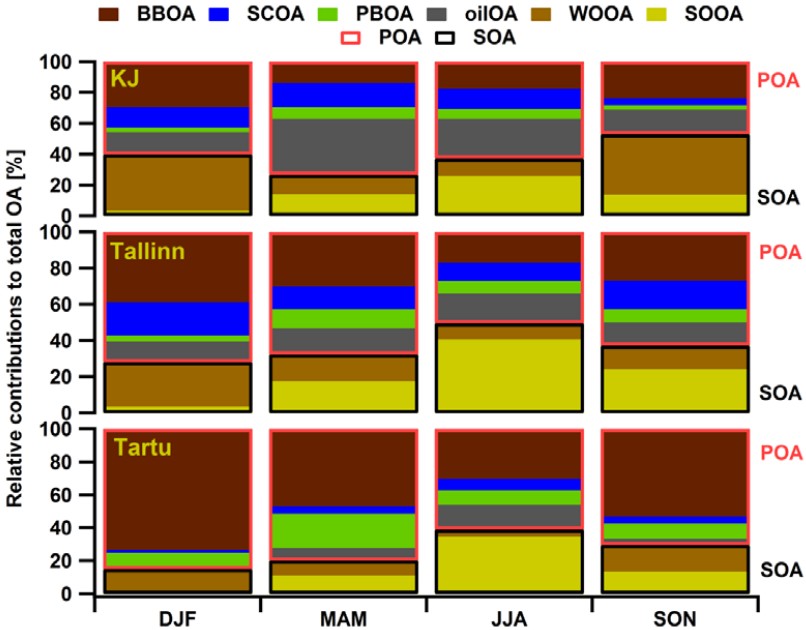

Figure 10. Seasonally averaged relative contributions of each factor to the total OA per site. The red and black boxes indicate the contributions of primary and secondary OA, respectively.

