# Peer review of "Development of a versatile source apportionment analysis based on positive matrix factorization: a case study of the seasonal variation of organic aerosol sources in Estonia."

_Atmospheric Chemistry and Physics, 2018_

## Referee Comment (RC1) · Anonymous Referee #1 · 11 Dec 2018

This article presents source apportionment of OA contained in PM10 in Estonia using PMF technique on off-line AMS data sets. It also estimate the uncertainty of the PMF solution by a bootstrap analysis. Overall, this manuscript is well written, and the findings of the study are in agreement with previous studies. I have minor concerns for the manuscript as listed in the following: 1. Number of samples: The author collected only 39 samples from KJ. I am afraid that only 10 samples/ season is not enough to present for the seasonal variation of organic aerosols. It would be more useful if the author could describe in more details of sampling time for 150 samples. Do those

samples collected from three sites in same sampling days/ time? 2. PMF techniques: The bootstrap techniques were commonly used to check uncertainty of PMF solutions for decades. I do not think it is "novel technique" as the author state in Page 3, Line 10. Could the author explain more the novelty of their bootstrap techniques? 3. In equation 3, do the authors assume the EC/WSOA ratio for each source is constant? If yes, please provide any references for that. In addition, the author should provide the correlation between measured EC and modeled EC concentration based on equation (3).
* * *

---

## Referee Comment (RC2) · Anonymous Referee #2 · 21 Dec 2018

General Comments: This manuscript investigates a year of organic aerosol in PM10 at three sites in Estonia. The organic aerosol was characterized by analyzing the water-soluble filter extract with an HR-ToF-AMS. The PM10 samples were 24-hour integrated PM10 quartz fiber filter samples using a high-volume sampler. Source apportionment of the AMS mass spectra was performed using positive matrix factorization. Seven factors were found between the three sites including an oil factor, sulfate-rich factor, summer oxygenated OA, winter oxygenated OA, dust, primary biological OA, and biomass burning OA. The BBOA, PBOA, WOOA, and SOOA factors were all validated with

externally measured organic markers. The dust factors were validated with external measurements of Ca2+. A bootstrap analysis is used to analyze the results and factor uncertainties. The paper thorough in its investigation and very well written. With some minor changes I suggest this paper be accepted.

Specific Comments: 4.1. For the estimation of traffic contribution to EC calculation what are the EC/WSOA values used? The ratio of EC/OC measured in biomass burning emissions highly variable (Pokhrel et al., 2016), as is SOA formation in biomass burning plumes (Jolleys et al., 2012; May et al., 2015). Given this range of possible EC/WSOA values, language as to uncertainties regarding this analysis should be added.

4.4. How many samples per season per site were there?

Fig 7. The colors of the two bars are very similar and hard to distinguish.

---

## Author Comment (AC1) · 21 Mar 2019

**Development of a versatile source apportionment analysis based on positive matrix factorization: a case study of the seasonal variation of organic aerosol sources in Estonia**

Athanasia Vlachou[1], Anna Tobler[1], Houssni Lamkaddam[1], Francesco Canonaco[1], Kaspar R. Daellenbach[1,a], Jean-Luc Jaffrezo[2], María Cruz Minguillón[3], Marek Maasikmets[4], Erik Teinemaa[4], Urs Baltensperger[1], Imad El Haddad[1], and André S. H. Prévôt[1]

[1]Department of General Energy Research, Paul Scherrer Institute, Villigen PSI, CH-5232, Switzerland

[2]Université Grenoble Alpes, CNRS, IRD, G-INP, IGE, 38000 Grenoble, France

[3]Institute of Environmental Assessment and Water Research (IDAEA), CSIC, 08034 Barcelona, Spain

[4]Estonian Environmental Research Centre, 10617, Tallinn, Estonia

[a]now at: Institute for Atmospheric and Earth System Research/Physics, Faculty of Science, University of Helsinki, P.O. Box 64, 00014, Helsinki, Finland

**Author's response:**

We thank Referee #1 for the careful revision and comments which helped improving the overall quality of the manuscript. A point-by-point answer (in regular typeset) to the referees' remarks (in the *italic typeset*) follows. Changes to the manuscript are indicated in blue font.

In the following page and lines references refer to the manuscript version reviewed by anonymous Referee #1.

1) *This article presents source apportionment of OA contained in PM10 in Estonia using PMF technique on off-line AMS data sets. It also estimate the uncertainty of the PMF solution by a bootstrap analysis. Overall, this manuscript is well written, and the findings of the study are in agreement with previous studies. I have minor concerns for the manuscript as listed in the following:*
*Number of samples: The author collected only 39 samples from KJ. I am afraid that only 10 samples/ season is not enough to present for the seasonal variation of organic aerosols. It would be more useful if the author could describe in more details of sampling time for 150 samples. Do those samples collected from three sites in same sampling days/ time?*

We agree with anonymous referee#1 that 39 samples are in general not enough to perform a robust PMF analysis. However, these 39 samples refer only to those from KJ and the PMF analysis was performed on 150 samples collected from three cities during the same period, where aerosol sources are very similar (except for the enhancement in OilOA in KJ). In addition, the errors on the PMF time-series estimated by the bootstrap analysis are similar at the three different locations (Table S3).

Aerosols were sampled for 24h (Page 3, line 32) at the three sites. The samples were not collected on the exact same dates at the 3 sites. Following the reviewer comment, the dates are now displayed in the following

Fig. S1 and Table S1, added in the supplementary. According to the suggestions of anonymous referee#1 we also changed the text in the Methods (Page 3, Line 33) as follows:

The measurements were performed with 24 h integrated $PM_{10}$ quartz fibre filter samples from KJ (31/08/2013 to 25/08/2014), Tallinn (05/09/2013 to 01/09/2014) and Tartu (05/09/2013 to 31/08/2014) (see Table S1, Fig. S1 for details).

[Figure]

Figure S1. Dates per site. KJ represented in red, Tallinn in green and Tartu in blue.

Table S1. Dates per site.

| KJ | Tallinn | Tartu | KJ | Tallinn | Tartu |
|---|---|---|---|---|---|
| **2013** | | | **2014** | | |
| 31.08.2013 | | | | | 03.01.2014 |
| | 05.09.2013 | 05.09.2013 | | 05.01.2014 | |
| 08.09.2013 | 08.09.2013 | | 06.01.2014 | | |
| | | 09.09.2013 | | | 07.01.2014 |
| | 12.09.2013 | | | 11.01.2014 | |
| | 13.09.2013 | | | 19.01.2014 | 19.01.2014 |
| | 16.09.2013 | | 22.01.2014 | | |
| | | 17.09.2013 | | 26.01.2014 | |
| 20.09.2013 | 20.09.2013 | | | 02.02.2014 | |
| | 22.09.2013 | | | | 27.01.2014 |
| | | 29.09.2013 | 03.02.2014 | 03.02.2014 | |
| | 30.09.2013 | | | 06.02.2014 | |
| 02.10.2013 | | | | 07.02.2014 | |
| | 07.10.2013 | | | | 08.02.2014 |
| | 10.10.2013 | | | 09.02.2014 | |
| | | 11.10.2013 | 11.02.2014 | | |
| 14.10.2013 | 14.10.2013 | | 15.02.2014 | | |
| 18.10.2013 | | | | 16.02.2014 | |
| | 21.10.2013 | | | | 20.02.2014 |
| | | 23.10.2013 | | 23.02.2014 | |
| 26.10.2013 | | | | | 24.02.2014 |
| | 27.10.2013 | 27.10.2013 | 27.02.2014 | | |
| | 04.11.2013 | | | 28.02.2014 | |

| | | | | | |
|---|---|---|---|---|---|
| 07.11.2013 | | | | 01.03.2014 | |
| | | 08.11.2013 | | 02.03.2014 | |
| | 10.11.2013 | | | | 04.03.2014 |
| | 17.11.2013 | | | | 08.03.2014 |
| 19.11.2013 | | | | 10.03.2014 | |
| | | 20.11.2013 | 11.03.2014 | | |
| 23.11.2013 | | | | | 16.03.2014 |
| | 24.11.2013 | | | 17.03.2014 | |
| | | 28.11.2013 | | | 20.03.2014 |
| 01.12.2013 | 01.12.2013 | | | 22.03.2014 | |
| | | | 23.03.2014 | | |
| | 08.12.2013 | | | 27.03.2014 | |
| 13.12.2013 | | | | | 28.03.2014 |
| | | 14.12.2013 | | 30.03.2014 | |
| | 16.12.2013 | | 31.03.2014 | | |
| 17.12.2013 | | | 04.04.2014 | | |
| | 23.12.2013 | | | 06.04.2014 | |
| | 26.12.2013 | 26.12.2013 | | | 09.04.2014 |
| 29.12.2013 | | | | 13.04.2014 | |
| | 30.12.2013 | | 16.04.2014 | | |
| | | | | 20.04.2014 | |
| | | | | | 21.04.2014 |
| | | | | 25.04.2014 | 25.04.2014 |
| | | | 28.04.2014 | | |
| | | | | 03.05.2014 | 03.05.2014 |
| | | | 06.05.2014 | | |
| | | | | 11.05.2014 | |
| | | | 12.05.2014 | | |
| | | | | | 15.05.2014 |
| | | | | 23.05.2014 | 23.05.2014 |
| | | | 24.05.2014 | 24.05.2014 | |
| | | | | 25.05.2014 | |
| | | | | | 27.05.2014 |
| | | | | 29.05.2014 | |
| | | | | 01.06.2014 | |
| | | | | | 04.06.2014 |
| | | | 05.06.2014 | 05.06.2014 | |
| | | | | 06.06.2014 | |
| | | | | 08.06.2014 | |
| | | | | 15.06.2014 | |
| | | | | | 16.06.2014 |
| | | | 17.06.2014 | | |
| | | | | | 20.06.2014 |
| | | | | 22.06.2014 | |
| | | | | | 28.06.2014 |
| | | | 29.06.2014 | 29.06.2014 | |
| | | | 07.07.2014 | 07.07.2014 | |
| | | | | | 10.07.2014 |
| | | | 12.07.2014 | | |
| | | | | 14.07.2014 | |
| | | | 20.07.2014 | | |

| | | |
|---|---|---|
| | | 22.07.2014 |
| | 25.07.2014 | |
| | 28.07.2014 | |
| | 31.07.2014 | |
| 01.08.2014 | | |
| | | 03.08.2014 |
| | 04.08.2014 | |
| 05.08.2014 | | |
| | | 07.08.2014 |
| | 11.08.2014 | |
| 13.08.2014 | | |
| | | 15.08.2014 |
| | 18.08.2014 | |
| | 24.08.2014 | |
| 25.08.2014 | | |
| | | 27.08.2014 |
| | | 31.08.2014 |
| | 01.09.2014 | |

2) *PMF techniques: The bootstrap techniques were commonly used to check uncertainty of PMF solutions for decades. I do not think it is "novel technique" as the author state in Page 3, Line 10. Could the author explain more the novelty of their bootstrap techniques?*

We did not intend to claim that the bootstrap technique used here for the uncertainty exploration of PMF is novel. We apologize for this misunderstanding. The novelty corresponds to the selection technique applied after the bootstrap analysis, which enables evaluating the quality of a large set of solutions. We changed the text accordingly (Page 3, Line 10):

In this study, we propose a novel technique to evaluate the selection of the PMF solutions generated through a large number of bootstrap iterations.

3) *In equation 3, do the authors assume the EC/WSOA ratio for each source is constant? If yes, please provide any references for that. In addition, the author should provide the correlation between measured EC and modeled EC concentration based on equation (3).*

According to the definition of PMF (Paatero, 1999), the sources are represented by constant profiles and varying intensities; therefore, as noted by the reviewer the EC/WSOA ratio for each source is constant. We note that this ratio is not assumed a priori, but determined through the PMF analysis. We also note that the PMF analysis performed to assess the EC sources is fully constrained. Therefore, EC modelled corresponds to the measured EC as shown by the figure below:

[Figure]

Correlation between EC measured by the thermal/optical method and the modelled EC calculated by PMF (EC = $EC_{tr} + EC_{bb} + EC_{oil} + EC_{rrd}$).

---

## Author Comment (AC2) · 21 Mar 2019

**Development of a versatile source apportionment analysis based on positive matrix factorization: a case study of the seasonal variation of organic aerosol sources in Estonia**

Athanasia Vlachou[1], Anna Tobler[1], Houssni Lamkaddam[1], Francesco Canonaco[1], Kaspar R. Daellenbach[1,a], Jean-Luc Jaffrezo[2], María Cruz Minguillón[3], Marek Maasikmets[4], Erik Teinemaa[4], Urs Baltensperger[1], Imad El Haddad[1], and André S. H. Prévôt[1]

[1]Department of General Energy Research, Paul Scherrer Institute, Villigen PSI, CH-5232, Switzerland

[2]Université Grenoble Alpes, CNRS, IRD, G-INP, IGE, 38000 Grenoble, France

[3]Institute of Environmental Assessment and Water Research (IDAEA), CSIC, 08034 Barcelona, Spain

[4]Estonian Environmental Research Centre, 10617, Tallinn, Estonia

[a]now at: Institute for Atmospheric and Earth System Research/Physics, Faculty of Science, University of Helsinki, P.O. Box 64, 00014, Helsinki, Finland

**Author's response:**

We thank Referee #2 for the careful revision and comments which helped improving the overall quality of the manuscript. A point-by-point answer (in regular typeset) to the referees' remarks (in the *italic typeset*) follows. Changes to the manuscript are indicated in blue font.

In the following page and lines references refer to the manuscript version reviewed by anonymous Referee #2.

1) *This manuscript investigates a year of organic aerosol in PM10 at three sites in Estonia. The organic aerosol was characterized by analyzing the watersoluble filter extract with an HR-ToF-AMS. The PM10 samples were 24-hour integrated PM10 quartz fiber filter samples using a high-volume sampler. Source apportionment of the AMS mass spectra was performed using positive matrix factorization. Seven factors were found between the three sites including an oil factor, sulfate-rich factor, summer oxygenated OA, winter oxygenated OA, dust, primary biological OA, and biomass burning OA. The BBOA, PBOA, WOOA, and SOOA factors were all validated with externally measured organic markers. The dust factors were validated with external measurements of Ca2+. A bootstrap analysis is used to analyze the results and factor u ncertainties. The paper thorough in its investigation and very well written. With some minor changes I suggest this paper be accepted.*

*Specific comment: For the estimation of traffic contribution to EC calculation what are the EC/WSOA values used? The ratio of EC/OC measured in biomass burning emissions highly variable (Pokhrel et al., 2016), as is SOA formation in biomass burning plumes (Jolleys et al., 2012; May et al., 2015). Given this range of possible EC/WSOA values, language as to uncertainties regarding this analysis should be added.*

We need to add some clarifications about the PMF analysis performed to apportion the EC contribution to the different sources. The analysis includes 4 variables: the time-series of the concentrations of EC, water soluble BBOA (WSBBOA), WSSCOA, and WSOilOA. Four factors are considered only

representing primary anthropogenic sources: traffic, biomass burning, road dust resuspension/tire-wear, and oil related processes. The contribution of EC in all profiles is not constrained (i.e. in none of the profiles is the EC/OA ratio fixed a priori). The contribution of the water-soluble organic aerosol from a certain source is also not constrained in the factor profile representing the source in question, while the organic aerosol from all other sources are set to 0 in this profile. As mentioned in the response to reviewer#1, such setting means that the EC variability is fully explained. In such setting, we do not assume any EC/WSOA in the traffic profile. As mentioned above, we also do not include any factor representing secondary OA. Therefore, uncertainties related to the EC/WSOA in aged biomass burning do not affect the analysis. We indeed agree with the reviewer comment that WSOA/EC is highly variable in primary sources, e.g. biomass burning. Therefore, we have performed a bootstrap analysis to assess the uncertainties in retrieving the contributions of the different sources to EC. These are represented as PDFs in Figure S4.

We have adapted the text in the manuscript as follows (Page 7, line 11):

Here, $EC_{bb}$, $EC_{oil}$ and $EC_{rrd}$ represent the EC concentration time-series related to the primary sources biomass burning, oil shale industry and resuspension of road dust/tire wear, respectively, while, a, b, and c are the EC/WSOA ratios characteristic of the emissions from the same sources and were obtained as outputs of the model. This new methodology, based on PMF, is especially pertinent as unlike other inversion techniques it sets positive constraints on a, b and c and offers the possibility of resolving the contributions of factors for which no constraints are available, here $EC_{tr}$.

We found that $EC_{tr}$ contributed 57% ± 5% to the total EC (on a yearly average), while 36% ± 5% of EC was attributed to biomass burning, 4% ± 1% to road dust resuspension and 3% ± 1% to the oil shale emissions (Fig. S4). The contribution of EC from fossil fuel combustion ($EC_{ff}$) measured at a site similar to Tartu, i.e. an Alpine valley in Southern Switzerland, Magadino in 2014 (Vlachou et al., 2018) was in agreement with our $EC_{tr}$ contribution, with a yearly average of 55% ± 7%. Also in Zurich, an urban site, $EC_{ff}$ ranged from 40% to 55% during winter 2012 (Zotter et al., 2014). From the $EC_{tr}$ contribution, we estimated that the HOC (obtained by multiplication of the $EC_{tr}$ time series with the HOC:EC ratio) contributed 4% to the total OC on a yearly average.

[Figure]

Figure S4. Probability density functions for the EC:WSOA ratios (*a* for WSBBOA, *b* for WSSCOA and *c* for WSoilOA) characteristic of the emissions from the same sources obtained by the 1000 PMF runs.

*2) How many samples per season per site were there?*

We gathered all the information in the following table. We will include it in the supplementary along with Table S1 and Fig. S1 which contain all the dates per site (as mentioned in the response to reviewer#1).

Table S2. Number of samples per season per site.

| | Number of samples per season | | |
|---|---|---|---|
| **Seasons** | **KJ** | **Tallinn** | **Tartu** |
| Summer | 11 | 16 | 11 |
| Autumn | 9 | 18 | 10 |
| Winter | 10 | 18 | 9 |
| Spring | 9 | 17 | 12 |

*3) Fig 7. The colours of the two bars are very similar and hard to distinguish.*

We agree with anonymous referee#2 and therefore we changed the colours from light and dark green to red and blue as shown in the plot below.

[Figure]

Figure 7. Examples of fumigated and non-fumigated mass spectra from two samples, with a high (KJ 05/06/2014, a), and a low dust concentration (Tallinn 19/01/2014, b).